# A SPHERICAL ANALYSIS OF ADAM WITH BATCH NORMALIZATION

Batch Normalization (BN) is a prominent deep learning technique. In spite of its apparent simplicity, its implications over optimization are yet to be fully understood. While previous studies mostly focus on the interaction between BN and stochastic gradient descent (SGD), we develop a geometric perspective which allows us to precisely characterize the relation between BN and Adam. More precisely, we leverage the radial invariance of groups of parameters, such as filters for convolutional neural networks, to translate the optimization steps on the $L_2$ unit hypersphere. This formulation and the associated geometric interpretation shed new light on the training dynamics. Firstly, we use it to derive the first effective learning rate expression of Adam. Then we show that, in the presence of BN layers, performing SGD alone is actually equivalent to a variant of Adam constrained to the unit hypersphere. Finally, our analysis outlines phenomena that previous variants of Adam act on and we experimentally validate their importance in the optimization process.

## 1 INTRODUCTION

The optimization process of deep neural networks is still poorly understood. Their training involves minimizing a high-dimensional non-convex function, which has been proved to be a NP-hard problem (Blum & Rivest, 1989). Yet, elementary gradient-based methods show good results in practice. To improve the quality of reached minima, numerous methods have stemmed in the last years and become common practices. One of the most prominent is Batch Normalization (BN) (Ioffe & Szegedy, 2015), which improves significantly both the optimization stability and the prediction performance; it is now used in most deep learning architectures. However, the interaction of BN with optimization and its link to regularization remain open research topics. Previous studies highlighted mechanisms of the interaction between BN and SGD, both empirically (Santurkar et al., 2018) and theoretically (Arora et al., 2019; Bjorck

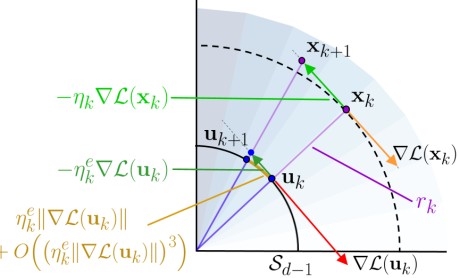

Figure 1: **Illustration of the spherical perspective for SGD.** The loss function $\mathcal{L}$ of a NN w.r.t. the parameters $\mathbf{x}_k \in \mathbb{R}^d$ of a neuron followed by a BN is radially invariant. The neuron update $\mathbf{x}_k \rightarrow \mathbf{x}_{k+1}$ in the original space, with velocity $\eta_k \nabla \mathcal{L}(\mathbf{x}_k)$, corresponds to an update $\mathbf{u}_k \rightarrow \mathbf{u}_{k+1}$ of its projection through an exponential map on the unit hypersphere $\mathcal{S}_{d-1}$ with velocity $\eta_k^e \|\nabla \mathcal{L}(\mathbf{u}_k)\|$ at order 2 (see details in Section 2.3).

et al., 2018; Hoffer et al., 2018b). None of them studied the interaction between BN and one of the most common adaptive schemes for Neural Networks (NN), Adam (Kingma & Ba, 2015), except van Laarhoven (2017), which tackled it only in the asymptotic regime. In this work, we provide an extensive analysis of the relation between BN and Adam during the whole training procedure.

One of the key effects of BN is to make NNs invariant to positive scalings of groups of parameters. The core idea of this paper is precisely to focus on these groups of radially-invariant parameters and analyze their optimization projected on the $L_2$ unit hypersphere (see Fig. 1), which is topologically equivalent to the quotient manifold of the parameter space by the scaling action. One could directly optimize parameters on the hypersphere as Cho & Lee (2017), yet, most optimization methods are still performed successfully in the original parameter space. Here we propose to study an optimization scheme for a given group of radially-invariant parameters through its image scheme on the unit hypersphere. This geometric perspective sheds light on the interaction between normalization layers and Adam, and also outlines an interesting link between standard SGD and a variant of Adam adapted and constrained to the unit hypersphere: AdamG (Cho & Lee, 2017). We believe this kind of analysis

is an important step towards a better understanding of the effect of BN on NN optimization. Please note that, although our discussion and experiments focus on BN, our analysis could be applied to any radially-invariant model.

The paper is organized as follows. In **Section 2**, we introduce our spherical framework to study the optimization of radially-invariant models. We also define a generic optimization scheme that encompasses methods such as SGD with momentum (SGD-M) and Adam. We then derive its image step on the unit hypersphere, leading to definitions and expressions of *effective learning rate* and *effective learning direction*. This new definition is explicit and has a clear interpretation, whereas the definition of van Laarhoven (2017) is asymptotic and the definitions of Arora et al. (2019) and of Hoffer et al. (2018b) are variational. In **Section 3**, we leverage the tools of our spherical framework to demonstrate that in presence of BN layers, SGD has an adaptive behaviour. Formally, we show that SGD is equivalent to AdamG, a variant of Adam adapted and constrained to the hypersphere, without momentum. In **Section 4**, we analyze the effective learning direction for Adam. The spherical framework highlights phenomena that previous variants of Adam (Loshchilov & Hutter, 2017; Cho & Lee, 2017) act on. We perform an empirical study of these phenomena and show that they play a significant role in the training of convolutional neural networks (CNNs). In **Section 5**, these results are put in perspective with related work.

Our main contributions are the following:
- A framework to analyze and compare order-1 optimization schemes of radially-invariant models;
- The first explicit expression of the effective learning rate for Adam;
- The demonstration that, in the presence of BN layers, standard SGD has an adaptive behaviour;
- The identification and study of geometrical phenomena that occur with Adam and impact significantly the training of CNNs with BN.

## 2 SPHERICAL FRAMEWORK AND EFFECTIVE LEARNING RATE

In this section, we provide background on radial invariance and introduce a generic optimization scheme.

Projecting the scheme update on the unit hypersphere leads to the formal definitions of effective learning rate and learning direction. This geometric perspective leads to the first explicit expression of the effective learning rate for Adam. The main notations are summarized in Figure 1.

### 2.1 RADIAL INVARIANCE

We consider a family of parametric functions $\phi_{\mathbf{x}} : \mathbb{R}^{in} \to \mathbb{R}^{out}$ parameterized by a group of radially-invariant parameters $\mathbf{x} \in \mathbb{R}^d \smallsetminus \{\mathbf{0}\}$, i.e., $\forall \rho > 0, \phi_{\rho \mathbf{x}} = \phi_{\mathbf{x}}$ (possible other parameters of $\phi_{\mathbf{x}}$ are omitted for clarity), a dataset $\mathcal{D} \subset \mathbb{R}^{in} \times \mathbb{R}^{out}$, a loss function $\ell : \mathbb{R}^{out} \times \mathbb{R}^{out} \to \mathbb{R}$ and a training loss function $\mathcal{L} : \mathbb{R}^d \to \mathbb{R}$ defined as:

$$\mathcal{L}(\mathbf{x}) \stackrel{\text{def}}{=} \frac{1}{|\mathcal{D}|} \sum_{(\mathbf{s}, \mathbf{t}) \in \mathcal{D}} \ell(\phi_{\mathbf{x}}(\mathbf{s}), \mathbf{t}). \tag{1}$$

It verifies: $\forall \rho > 0, \mathcal{L}(\rho \mathbf{x}) = \mathcal{L}(\mathbf{x})$. In the context of NNs, the group of radially-invariant parameters $\mathbf{x}$ can be the parameters of a single neuron in a linear layer or the parameters of a whole filter in a convolutional layer, followed by BN (see Appendix A for details, and Appendix B for the application to other normalization schemes such as InstanceNorm (Ulyanov et al., 2016), LayerNorm (Ba et al., 2016) or GroupNorm (Wu & He, 2018)).

The quotient of the parameter space by the equivalence relation associated to radial invariance is topologically equivalent to a sphere. We consider here the $L_2$ sphere $\mathcal{S}_{d-1} = \{\mathbf{u} \in \mathbb{R}^d / \|\mathbf{u}\|_2 = 1\}$ whose canonical metric corresponds to angles: $d_{\mathcal{S}}(\mathbf{u}_1, \mathbf{u}_2) = \arccos(\langle \mathbf{u}_1, \mathbf{u}_2 \rangle)$. This choice of metric is relevant to study NNs since filters in CNNs or neurons in MLPs are applied through scalar product to input data. Besides, normalization in BN layers is also performed using the $L_2$ norm.

Our framework relies on the decomposition of vectors into radial and tangential components. During optimization, we write the radially-invariant parameters at step $k \geq 0$ as $\mathbf{x}_k = r_k \mathbf{u}_k$ where $r_k = \|\mathbf{x}_k\|$ and $\mathbf{u}_k = \mathbf{x}_k / \|\mathbf{x}_k\|$. For any quantity $\mathbf{q}_k \in \mathbb{R}^d$ at step $k$, we write $\mathbf{q}_k^{\perp} = \mathbf{q}_k - \langle \mathbf{q}_k, \mathbf{u}_k \rangle \mathbf{u}_k$ its tangential component relatively to the current direction $\mathbf{u}_k$.

The following lemma states that the gradient of a radially-invariant loss function is tangential and $-1$ homogeneous:

**Lemma 1** (Gradient of a function with radial invariance). *If $\mathcal{L} : \mathbb{R}^d \to \mathbb{R}$ is radially invariant and almost everywhere differentiable, then, for all $\rho > 0$ and all $\mathbf{x} \in \mathbb{R}^d$ where $\mathcal{L}$ is differentiable:*

$$\langle \nabla \mathcal{L}(\mathbf{x}), \mathbf{x} \rangle = 0 \quad \text{and} \quad \nabla \mathcal{L}(\mathbf{x}) = \rho \, \nabla \mathcal{L}(\rho \mathbf{x}). \tag{2}$$

## 2.2 GENERIC OPTIMIZATION SCHEME

There is a large body of literature on optimization schemes (Sutskever et al., 2013; Duchi et al., 2011; Tieleman & Hinton, 2012; Kingma & Ba, 2015; Loshchilov & Hutter, 2019). We focus here on two of the most popular ones, namely SGD and Adam (Kingma & Ba, 2015). Yet, to establish general results that may apply to a variety of other schemes, we introduce here a *generic optimization update*:

$$\mathbf{x}_{k+1} = \mathbf{x}_k - \eta_k \mathbf{a}_k \oslash \mathbf{b}_k, \tag{3}$$
$$\mathbf{a}_k = \beta \mathbf{a}_{k-1} + \nabla \mathcal{L}(\mathbf{x}_k) + \lambda \mathbf{x}_k, \tag{4}$$

where $\mathbf{x}_k \in \mathbb{R}^d$ is the group of radially-invariant parameters at iteration $k$, $\mathcal{L}$ is the group's loss estimated on a batch of input data, $\mathbf{a}_k \in \mathbb{R}^d$ is a momentum, $\mathbf{b}_k \in \mathbb{R}^d$ is a division vector that can depend on the trajectory $(\mathbf{x}_i, \nabla \mathcal{L}(\mathbf{x}_i))_{i \in [\![0,k]\!]}$, $\eta_k \in \mathbb{R}$ is the scheduled trajectory-independent learning rate, $\oslash$ denotes the Hadamard element-wise division, $\beta$ is the momentum parameter, and $\lambda$ is the $L_2$-regularization parameter. We show how it encompasses several known optimization schemes.

*Stochastic gradient descent (SGD)* has proven to be an effective optimization method in deep learning. It can include $L_2$ regularization (also called weight decay) and momentum. Its updates are:

$$\mathbf{x}_{k+1} = \mathbf{x}_k - \eta_k \mathbf{m}_k, \tag{5}$$
$$\mathbf{m}_k = \beta \mathbf{m}_{k-1} + \nabla \mathcal{L}(\mathbf{x}_k) + \lambda \mathbf{x}_k, \tag{6}$$

where $\mathbf{m}_k$ is the momentum, $\beta$ is the momentum parameter, and $\lambda$ is the $L_2$-regularization parameter. It corresponds to our generic scheme (Eqs. 3-4) with $\mathbf{a}_k = \mathbf{m}_k$ and $\mathbf{b}_k = [1 \cdots 1]^\top$.

*Adam* is likely the most common adaptive scheme for NNs. Its updates are:

$$\mathbf{x}_{k+1} = \mathbf{x}_k - \eta_k \frac{\mathbf{m}_k}{1 - \beta_1^{k+1}} \oslash \sqrt{\frac{\mathbf{v}_k}{1 - \beta_2^{k+1}} + \epsilon}, \tag{7}$$

$$\mathbf{m}_k = \beta_1 \mathbf{m}_{k-1} + (1 - \beta_1)(\nabla \mathcal{L}(\mathbf{x}_k) + \lambda \mathbf{x}_k), \quad \mathbf{v}_k = \beta_2 \mathbf{v}_{k-1} + (1 - \beta_2)(\nabla \mathcal{L}(\mathbf{x}_k) + \lambda \mathbf{x}_k)^2, \tag{8}$$

where $\mathbf{m}_k$ is the momentum with parameter $\beta_1$, $\mathbf{v}_k$ is the second-order moment with parameter $\beta_2$, and $\epsilon$ prevents division by zero. (Here and in the following, the square and the square root of a vector are to be understood as element-wise.) It corresponds to our generic scheme (Eqs. 3-4) with $\beta = \beta_1$ and:

$$\mathbf{a}_k = \frac{\mathbf{m}_k}{1 - \beta_1}, \qquad \mathbf{b}_k = \frac{1 - \beta_1^{k+1}}{1 - \beta_1} \sqrt{\frac{\mathbf{v}_k}{1 - \beta_2^{k+1}} + \epsilon}. \tag{9}$$

## 2.3 IMAGE OPTIMIZATION ON THE HYPERSPHERE

The radial invariance implies that the radial part of the parameter update $\mathbf{x}$ does not change the function $\phi_\mathbf{x}$ encoded by the model, nor does it change the loss $\mathcal{L}(\mathbf{x})$. The goal of training is to find the best possible function encodable by the network. Due to radial invariance, the parameter space projected on the unit hypersphere is topologically closer to the functional space of the network than the full parameter space. It hints that looking at optimization behaviour on the unit hypersphere might be interesting. Thus, we need to separate the quantities that can (tangential part) and cannot (radial part) change the model function. Theorem 2 formulates the spherical decomposition (Eqs. 3-4) in simple terms. It relates the update of radially-invariant parameters in the parameter space $\mathbb{R}^d$ and their update on $\mathcal{S}_{d-1}$ through an exponential map.

**Theorem 2** (Image step on $\mathcal{S}_{d-1}$). *The update of a group of radially-invariant parameters $\mathbf{x}_k$ at step $k$ corresponds to an update of its projection $\mathbf{u}_k$ on $\mathcal{S}_{d-1}$ through an exponential map at $\mathbf{u}_k$ with velocity $\eta_k^e \mathbf{c}_k^\perp$, at order 3:*

$$\mathbf{u}_{k+1} = \mathrm{Exp}_{\mathbf{u}_k}\left( -\left[1 + O\left(\left(\eta_k^e \|\mathbf{c}_k^\perp\|\right)^2\right)\right] \eta_k^e \mathbf{c}_k^\perp \right), \tag{10}$$

where $\mathrm{Exp}_{\mathbf{u}_k}$ is the exponential map on $\mathcal{S}_{d-1}$, and with

$$\mathbf{c}_k \overset{\text{def}}{=} r_k \mathbf{a}_k \oslash \frac{\mathbf{b}_k}{d^{-1/2}\|\mathbf{b}_k\|}, \quad \eta_k^e \overset{\text{def}}{=} \frac{\eta_k}{r_k^2 d^{-1/2}\|\mathbf{b}_k\|} \left(1 - \frac{\eta_k \langle \mathbf{c}_k, \mathbf{u}_k \rangle}{r_k^2 d^{-1/2}\|\mathbf{b}_k\|}\right)^{-1}. \tag{11}$$

*More precisely:*

$$\mathbf{u}_{k+1} = \frac{\mathbf{u}_k - \eta_k^e \mathbf{c}_k^\perp}{\sqrt{1 + (\eta_k^e\|\mathbf{c}_k^\perp\|)^2}}. \tag{12}$$

The proof is given in Appendix C.1.1 and the theorem is illustrated in the case of SGD in Figure 1. Note that with typical values in CNN training we have $1 - \frac{\eta_k \langle \mathbf{c}_k, \mathbf{u}_k \rangle}{r_k^2 d^{-1/2}\|\mathbf{b}_k\|} > 0$, which is a property needed for the proof. Another hypothesis is that steps on the hypersphere are shorter than $\pi$. These hypotheses are discussed and empirically verified in Appendix C.1.2.

## 2.4 EFFECTIVE LEARNING RATE FOR ADAM

In Theorem 2, the normalized parameters update in Eq. 10 can be read $\mathbf{u}_{k+1} \approx \mathrm{Exp}_{\mathbf{u}_k}\left(-\eta_k^e \mathbf{c}_k^\perp\right)$, where $\eta_k^e$ and $\mathbf{c}_k^\perp$ can then be respectively interpreted as the learning rate and the direction of an optimization step constrained to $\mathcal{S}_{d-1}$ since $\mathbf{a}_k$ is the momentum and, with Lemma 1, the quantity $r_k \mathbf{a}_k$ in $\mathbf{c}_k$ can be seen as *a momentum on the hypersphere*. Due to the radial invariance, only the change of parameter on the unit hypersphere corresponds to a change of model function. Hence we can interpret $\eta_k^e$ and $\mathbf{c}_k^\perp$ as *effective learning rate* and *effective learning direction*. In other words, these quantities correspond to the learning rate and direction on the hypersphere that reproduce the function update of the optimization step.

Using Theorem 2, we can derive actual effective learning rates for any optimization scheme that fits our generic framework. These expressions, summarized in Table 1 are explicit and have a clear interpretation, in contrast to learning rates in (van Laarhoven, 2017), which are approximate and asymptotic, and in (Hoffer et al., 2018a; Arora et al., 2019), which are variational and restricted to SGD without momentum only.

In particular, we provide the first explicit expression of the effective learning rate for Adam:

$$\eta_k^e = \frac{\eta_k}{r\nu_k}\left(1 - \frac{\eta_k\langle\mathbf{c}_k, \mathbf{u}_k\rangle}{r\nu_k}\right)^{-1} \tag{13}$$

**Table 1:** Effective learning rate and direction for optimization schemes ($k$ omitted), with $\nu = rd^{-1/2}\|\mathbf{b}\|$.

| Scheme | $\eta^e$ | $\mathbf{c}^\perp$ |
|---|---|---|
| SGD | $\frac{\eta}{r^2}$ | $\nabla\mathcal{L}(\mathbf{u})$ |
| SGD + $L_2$ | $\frac{\eta}{r^2(1-\eta\lambda)}$ | $\nabla\mathcal{L}(\mathbf{u})$ |
| SGD-M | $\frac{\eta}{r^2}(1 - \frac{\eta\langle\mathbf{c},\mathbf{u}\rangle}{r^2})^{-1}$ | $\mathbf{c}^\perp$ |
| Adam | $\frac{\eta}{r\nu}(1 - \frac{\eta\langle\mathbf{c},\mathbf{u}\rangle}{r\nu})^{-1}$ | $\mathbf{c}^\perp$ |

where $\nu_k = r_k d^{-1/2}\|\mathbf{b}_k\|$ is homogeneous to the norm of a gradient on the hypersphere and can be related to an *second-order moment on the hypersphere* (see Appendix.C.1.3 for details). This notation also simplifies the in-depth analysis in Section 4, allowing a better interpretation of formulas.

The expression of the effective learning rate of Adam, i.e., the amplitude of the step taken on the hypersphere, reveals a dependence on the dimension $d$ (through $\nu$) of the considered group of radially-invariant parameters. In the case of an MLP or CNN that stacks layers with neurons or filters of different dimensions, the learning rate is thus tuned differently from one layer to another.

We can also see that for all schemes the learning rate is tuned by the dynamics of radiuses $r_k$, which follow:

$$\frac{r_{k+1}}{r_k} = \left(1 - \frac{\eta_k\langle\mathbf{c}_k, \mathbf{u}_k\rangle}{r_k^2 d^{-1/2}\|\mathbf{b}_k\|}\right)\sqrt{1 + (\eta_k^e\|\mathbf{c}_k^\perp\|)^2}. \tag{14}$$

In contrast to previous studies (Arora et al., 2019; van Laarhoven, 2017), this result demonstrates that for momentum methods, $\langle\mathbf{c}_k, \mathbf{u}_k\rangle$, which involves accumulated gradients terms in the momentum as well as $L_2$ regularization, tunes the learning rate (*cf.* Fig.1).

## 3 SGD IS A VARIATION OF ADAM ON THE HYPERSPHERE

We leverage the tools introduced in the spherical framework to find a scheme constrained to the hypersphere that is equivalent to SGD. It shows that for radially-invariant models, SGD is actually an adaptive optimization method. Formally SGD is equivalent to a version of AdamG, a variation of Adam adapted and constrained to the unit hypersphere, without momentum.

### 3.1 EQUIVALENCE BETWEEN TWO OPTIMIZATION SCHEMES

Due to the radial invariance, the functional space of the model is encoded by $\mathcal{S}_{d-1}$. In other words, two schemes with the same sequence of groups of radially-invariant parameters on the hypersphere $(\mathbf{u}_k)_{k \geq 0}$ encode the same sequence of model functions. Two optimization schemes $S$ and $\tilde{S}$ are equivalent iff $\forall k \geq 0, \mathbf{u}_k = \tilde{\mathbf{u}}_k$. By using Eq. 12, we obtain the following lemma, which is useful to prove the equivalence of two given optimization schemes:

**Lemma 3** (Sufficient condition for the equivalence of optimization schemes).

$$\left\{ \begin{array}{l} \mathbf{u}_0 = \tilde{\mathbf{u}}_0 \\ \forall k \geq 0, \eta_k^e = \tilde{\eta}_k^e, \mathbf{c}_k^\perp = \tilde{\mathbf{c}}_k^\perp \end{array} \right. \Rightarrow \forall k \geq 0, \mathbf{u}_k = \tilde{\mathbf{u}}_k. \tag{15}$$

### 3.2 A HYPERSPHERE-CONSTRAINED SCHEME EQUIVALENT TO SGD

We now study, within our spherical framework, SGD with $L_2$ regularization, i.e., the update $\mathbf{x}_{k+1} = \mathbf{x}_k - \eta_k(\nabla\mathcal{L}(\mathbf{x}_k) - \lambda_k\mathbf{x}_k)$. From the effective learning rate expression, we know that SGD yields an adaptive behaviour because it is scheduled by the radius dynamic, which depends on gradients. In fact, the tools in our framework allow us to find a scheme constrained to the unit hypersphere that is equivalent to SGD: AdamG (Cho & Lee, 2017). More precisely, it is AdamG with a null momentum factor $\beta_1 = 0$, an non-null initial second-order moment $v_0$, an offset of the scalar second-order moment $k + 1 \to k$ and the absence of the bias correction term $1 - \beta_2^{k+1}$. Dubbed AdamG* this scheme reads:

$$(\text{AdamG*}) : \left\{ \begin{array}{l} \hat{\mathbf{x}}_{k+1} = \mathbf{x}_k - \eta_k \frac{\nabla\mathcal{L}(\mathbf{x}_k)}{\sqrt{v_k}}, \\ \mathbf{x}_{k+1} = \frac{\hat{\mathbf{x}}_{k+1}}{\|\hat{\mathbf{x}}_{k+1}\|}, \\ v_{k+1} = \beta v_k + \|\nabla\mathcal{L}(\mathbf{x}_k)\|^2. \end{array} \right.$$

Starting from SGD, we first use Lemma 3 to find an equivalence scheme with simpler radius dynamic. We resolve this radius dynamic with a Taylor expansion at order 2 in $(\eta_k\|\nabla\mathcal{L}(\mathbf{u}_k)\|)^2/r_k^2$. A second use of Lemma 3 finally leads to the following scheme equivalence in Theorem (see proof in Appendix C.1.4). If we call « equivalent at order 2 in the step » a scheme equivalence that holds when we use for $r_k$ an expression that satisfies the radius dynamic with a Taylor expansion at order 2 we have the following theorem:

**Theorem 4** (SGD equivalent scheme on the unit hypersphere). *For any $\lambda > 0, \eta > 0, r_0 > 0$, we have the following equivalence when using the radius dynamic at order 2 in $(\eta_k\|\nabla\mathcal{L}(\mathbf{u}_k)\|)^2/r_k^2$:*

$$\left\{ \begin{array}{l} (\text{SGD}) \\ \mathbf{x}_0 = r_0\mathbf{u}_0 \\ \lambda_k = \lambda \\ \eta_k = \eta \end{array} \right. \text{ is scheme-equivalent at order 2 in step with } \left\{ \begin{array}{l} (\text{AdamG*}) \\ \mathbf{x}_0 = \mathbf{u}_0 \\ \beta = (1 - \eta\lambda)^4 \\ \eta_k = (2\beta)^{-1/2} \\ v_0 = r_0^4(2\eta^2\beta^{1/2})^{-1}. \end{array} \right.$$

This result is unexpected because SGD, which is not adaptive by itself, is equivalent to a second order moment adaptive method The scheduling performed by the radius dynamics actually replicates the effect of dividing the learning rate by the second-order moment of the gradient norm: $v_k$. First, the only assumption for this equivalence is to neglect the approximation in the Taylor expansion at order 2 of the radius which is highly verified in practice (order of magnitude of $1e - 4$ isee Appendix C.1.5). Second, with standard values of the hyper-parameters : learning rate $\eta < 1$ and weight decay $\lambda < 1$, we have $\beta \leq 1$ which corresponds to a standard value for a moment factor. Interestingly, the L2 regularization parameter $\lambda$ controls the memory of the past gradients norm. If $\beta = 1$ (with $\lambda = 0$) there is no attenuation, each gradient norm has the same contribution in the order of two moments. If $\lambda \neq 0$, there is a decay factor ($\beta < 1$) on past gradients norm in the order 2 moment.

## 4 GEOMETRIC PHENOMENA IN ADAM

Our framework with its geometrical interpretation reveals intriguing behaviors occurring in Adam. The unit hypersphere is enough to represent the functional space encoded by the network. From the perspective of manifold optimization, the optimization direction would only depend on the trajectory on that manifold. In the case of Adam, the effective direction not only depends on the trajectory on the hypersphere but also on the deformed gradients and additional radial terms. These terms are thus likely to play a role in Adam optimization.

In order to understand their role, we describe these geometrical phenomena in Section 4.1. Interestingly, previous variants of Adam, AdamW (Loshchilov & Hutter, 2017) and AdamG (Cho & Lee, 2017) are related to these phenomena. To study empirically their importance, we consider in Section 4.2 variants of Adam that first provide a direction intrinsic to the unit hypersphere, without deformation of the gradients, and then where radial terms are decoupled from the direction. The empirical study of these variants over a variety of datasets and architectures suggests that these behaviors do play a significant role in CNNs training with BN.

### 4.1 IDENTIFICATION OF GEOMETRICAL PHENOMENA IN ADAM

Here, we perform an in-depth analysis of the effective learning direction of Adam.

**(a) Deformed gradients.** Considering the quantities defined for a generic scheme in Eq. 11, $\mathbf{b}_k$ has a deformation effect on $\mathbf{a}_k$, due to the Hadamard division by $\frac{\mathbf{b}_k}{d^{-1/2}\|\mathbf{b}_k\|}$, and a scheduling effect $d^{-1/2}\|\mathbf{b}_k\|$ on the effective learning rate. In the case where the momentum factor is null $\beta_1 = 0$, the direction of the update at step $k$ is $\nabla\mathcal{L}(\mathbf{u}_k) \oslash \frac{\mathbf{b}_k}{d^{-1/2}\|\mathbf{b}_k\|}$ (Eq. 11) and the deformation $\frac{\mathbf{b}_k}{d^{-1/2}\|\mathbf{b}_k\|}$ may push the direction of the update outside the tangent space of $\mathcal{S}_{d-1}$ at $\mathbf{u}_k$, whereas the gradient itself lies in the tangent space. This deformation is in fact not isotropic: the displacement of the gradient from the tangent space depends on the position of $\mathbf{u}_k$ on the sphere. We illustrate this anisotropy in Fig. 2(b).

**(b) Additional radial terms.** In the momentum on the sphere $\mathbf{c}_k$, quantities that are radial (resp. orthogonal) at a point on the sphere may not be radial (resp. orthogonal) at another point. To clarify the contribution of $\mathbf{c}_k$ in the effective learning direction $\mathbf{c}_k^\perp$, we perform the following decomposition (*cf.* Appendix D.1):

$$\mathbf{c}_k = (\mathbf{c}_k^{\mathrm{grad}} + \lambda r_k^2 \mathbf{c}_k^{L_2}) \oslash \frac{\mathbf{b}_k}{d^{-1/2}\|\mathbf{b}_k\|} \quad \text{with:} \tag{16}$$

$$\mathbf{c}_k^{\mathrm{grad}} \overset{\text{def}}{=} \nabla\mathcal{L}(\mathbf{u}_k) + \sum_{i=0}^{k-1} \beta^{k-i} \frac{r_k}{r_i} \nabla\mathcal{L}(\mathbf{u}_i) \quad \text{and} \quad \mathbf{c}_k^{L_2} \overset{\text{def}}{=} \mathbf{u}_k + \sum_{i=0}^{k-1} \beta^{k-i} \frac{r_i}{r_k} \mathbf{u}_i. \tag{17}$$

**1. Contribution of $\mathbf{c}_k^{\mathrm{grad}}$.** At step $k$, the contribution of each past gradient corresponds to the orthogonal part $\nabla\mathcal{L}(\mathbf{u}_i) - \langle\nabla\mathcal{L}(\mathbf{u}_i), \mathbf{u}_k\rangle\mathbf{u}_k$. It impacts the effective learning direction depending on its orientation relatively to $\mathbf{u}_k$. Two past points, although equally distant from $\mathbf{u}_k$ on the sphere and with equal gradient amplitude may thus contribute differently in $\mathbf{c}_k^\perp$ due to their orientation (*cf.* Fig. 2(c)).

**2. Contribution of $\mathbf{c}_k^{L_2}$.** Naturally, the current point $\mathbf{u}_k$ does not contribute to the effective learning direction $\mathbf{c}_k^\perp$, unlike the history of points in $\sum_{i=0}^{k-1} \beta^{k-i} \frac{r_i}{r_k}\mathbf{u}_i$, which does. This dependency can be avoided if we decouple the $L_2$ regularization, in which case we do not accumulate $L_2$ terms in the momentum. This shows that the decoupling proposed in AdamW (Loshchilov & Hutter, 2019) actually removes the contribution of $L_2$ regularization in the effective learning direction.

**(c) The radius ratio** $\frac{r_k}{r_i}$ present in both $\mathbf{c}_k^{\mathrm{grad}}$ and $\mathbf{c}_k^{L_2}$ (in inverse proportion) impacts the effective learning direction $\mathbf{c}_k^\perp$: it can differ for identical sequences $(\mathbf{u}_i)_{i\leq k}$ on the sphere but with distinct radius histories $(r_i)_{i\leq k}$. Since the radius is closely related to the effective learning rate, it means that the effective learning direction $\mathbf{c}_k^\perp$ is adjusted according to the learning rates history.

Note that AdamG (Cho & Lee, 2017), by constraining the optimization to the unit hypersphere and thus removing $L_2$ regularization, neutralizes all the above phenomena. However, this method has no scheduling effect allowed by the radius dynamics (*cf.* Eq.14) since it is kept constant during training.

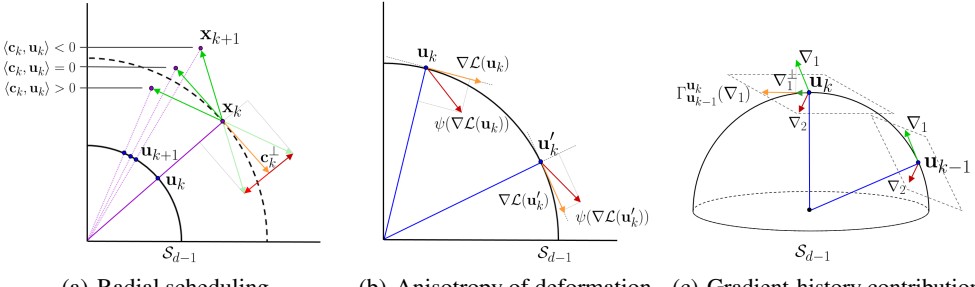

| (a) Radial scheduling | (b) Anisotropy of deformation | (c) Gradient-history contribution |

**Figure 2:** (a) Effect of the radial part of $\mathbf{c}_k$ on the displacement on $\mathcal{S}_{d-1}$; (b) Example of anisotropy and sign instability for the deformation $\psi(\nabla\mathcal{L}(\mathbf{u}_k)) = \nabla\mathcal{L}(\mathbf{u}_k) \oslash \frac{|\nabla\mathcal{L}(\mathbf{u}_k)|}{d^{-1/2}\|\nabla\mathcal{L}(\mathbf{u}_k)\|}$ (where $|\cdot|$ is the element-wise absolute value) occurring in Adam's first optimization step; (c) Different contribution in $\mathbf{c}_k^\perp$ of two past gradients $\nabla_1$ and $\nabla_2$ of equal norm, depending on their orientation. Illustration of the transport of $\nabla_1$ from $\mathbf{u}_{k-1}$ to $\mathbf{u}_k$ : $\Gamma_{\mathbf{u}_{k-1}}^{\mathbf{u}_k}(\nabla_1)$ (*cf.* Appendix D.2 for details)

## 4.2 Empirical study

To study empirically the importance of the identified geometric phenomena, we perform an ablation study: we compare the performance (accuracy and training loss speed) of Adam and variants that neutralize each of them. We recall that AdamW neutralizes **(b2)** and that AdamG neutralizes all of above phenomena but loses the scheduling effect identified in Eq. 14. To complete our analysis, we use geometrical tools to design variations of Adam which neutralizes sequentially each phenomenon while preserving the natural scheduling effect in Theorem 2. We neutralize **(a)** by replacing the element-wise second-order moment, **(b1)** and **(b2)** by transporting the momentum from a current point to the new one, **(c)** by re-scaling the momentum at step $k$. The details are in Appendix. D.2. The final scheme reads:

$$\mathbf{x}_{k+1} = \mathbf{x}_k - \eta_k \frac{\mathbf{m}_k}{1 - \beta_1^{k+1}} \Big/ \sqrt{\frac{v_k}{1 - \beta_2^{k+1}} + \epsilon}, \tag{18}$$

$$\mathbf{m}_k = \beta_1 \frac{r_{k-1}}{r_k} \Gamma_{\mathbf{u}_{k-1}}^{\mathbf{u}_k}(\mathbf{m}_{k-1}) + (1 - \beta_1)(\nabla\mathcal{L}(\mathbf{x}_k) + \lambda\mathbf{x}_k), \tag{19}$$

$$v_k = \beta_2 \frac{r_{k-1}^2}{r_k^2} v_{k-1} + (1 - \beta_2) d^{-1} \|\nabla\mathcal{L}(\mathbf{x}_k) + \lambda\mathbf{x}_k\|^2, \tag{20}$$

where $\Gamma_{\mathbf{u}_{k-1}}^{\mathbf{u}_k}$ is the hypersphere canonical transport from $\mathbf{u}_{k-1}$ to $\mathbf{u}_k$. Implementation details are in Appendix D.3.

**Protocol.** For evaluation, we conduct experiments on two architectures: VGG16 (Simonyan & Zisserman, 2015) and ResNet (He et al., 2016) – more precisely ResNet20, a simple variant designed for small images (He et al., 2016), and ResNet18, a popular variant for image classification. We consider three datasets: SVHN (Netzer et al., 2011), CIFAR10 and CIFAR100 (Krizhevsky et al., 2009).

Since our goal is to evaluate the significance of phenomena on radially-invariant parameters, i.e., the convolution filters followed by BN, we only apply variants of Adam including AdamG and AdamW on convolution layers. For comparison consistency, we keep standard Adam on the remaining parameters. We also use a fixed grid hyperparameter search budget and frequency for each method and each architecture (see Appendix D.3 for details).

**Results.** In Table 2 we report quantitative results of Adam variants across architectures and datasets. In addition, we compare the evolution of the training loss in Fig. 3. We observe that each phenomenon displays a specific trade-off between generalization (accuracy on the test set) and training speed, as following. Neutralizing **(a)** has little effect on the speed over Adam, yet achieves better accuracy. Although it slows down training, neutralizing **(ab)** leads to minima with the overall best accuracy on test set. Note that AdamW[†] neutralizes **(b2)** with its decoupling and is the fastest method, but finds minima with overall worst generalization properties. By constraining the optimization to the hypersphere, AdamG[†] speeds up training over the other variants. Finally, neutralizing **(c)** with Adam

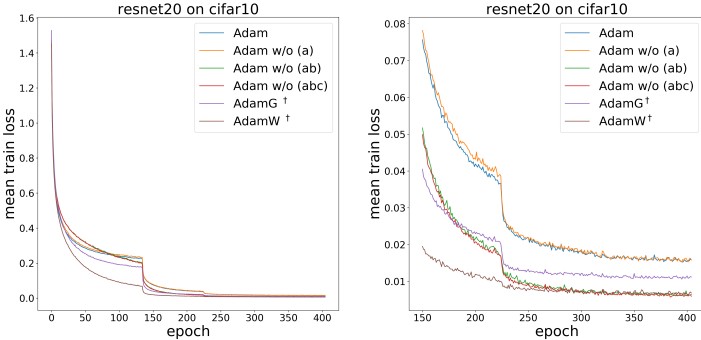

**Figure 3: Training speed comparison with ResNet20 on CIFAR10.** *Left:* Mean training loss over all training epochs (averaged across 5 seeds) for different Adam variants. *Right:* Zoom-in on the last epochs. Please refer to Table 2 for the corresponding accuracies.

w/o **(abc)** brings a slight acceleration, though reaches lower accuracy than Adam w/o **(ab)**. We can see that the revealed geometrical phenomena impact substantially training of BN-equipped CNNs.

**Table 2: Accuracy of Adam and its variants.** The figures in this table are the mean top1 accuracy $\pm$ the standard deviation over 5 seeds on the test set for CIFAR10, CIFAR100 and on the validation set for SVHN. [†] indicates that the original method is only used on convolutional filters while Adam is used for other parameters.

| Method | CIFAR10 | | | CIFAR100 | | SVHN | |
| | ResNet20 | ResNet18 | VGG16 | ResNet18 | VGG16 | ResNet18 | VGG16 |
|---|---|---|---|---|---|---|---|
| Adam | $90.98 \pm 0.06$ | $93.77 \pm 0.20$ | $92.83 \pm 0.17$ | $71.30 \pm 0.36$ | $68.43 \pm 0.16$ | $95.32 \pm 0.23$ | $95.57 \pm 0.20$ |
| AdamW[†] | $90.19 \pm 0.24$ | $93.61 \pm 0.12$ | $92.53 \pm 0.25$ | $67.39 \pm 0.27$ | $71.37 \pm 0.22$ | $95.13 \pm 0.15$ | $94.97 \pm 0.08$ |
| AdamG[†] | $91.64 \pm 0.17$ | $94.67 \pm 0.12$ | $93.41 \pm 0.17$ | $73.76 \pm 0.34$ | $70.17 \pm 0.20$ | $95.73 \pm 0.05$ | $95.70 \pm 0.25$ |
| Adam w/o (a) | $91.15 \pm 0.11$ | $93.95 \pm 0.23$ | $92.92 \pm 0.11$ | $74.44 \pm 0.22$ | $68.73 \pm 0.27$ | $95.75 \pm 0.09$ | $95.66 \pm 0.09$ |
| Adam w/o (ab) | $\mathbf{91.92 \pm 0.18}$ | $\mathbf{95.11 \pm 0.10}$ | $\mathbf{93.89 \pm 0.09}$ | $\mathbf{76.15 \pm 0.25}$ | $\mathbf{71.53 \pm 0.19}$ | $\mathbf{96.05 \pm 0.12}$ | $\mathbf{96.22 \pm 0.09}$ |
| Adam w/o (abc) | $91.81 \pm 0.20$ | $94.92 \pm 0.05$ | $93.75 \pm 0.06$ | $75.28 \pm 0.35$ | $71.45 \pm 0.13$ | $95.84 \pm 0.07$ | $95.82 \pm 0.05$ |

## 5 RELATED WORK

**Understanding Batch Normalization.** Albeit conceptually simple, BN has been shown to have complex implications over optimization. The argument of Internal Covariate Shift reduction (Ioffe & Szegedy, 2015) has been challenged and shown to be secondary to smoothing of optimization landscape (Santurkar et al., 2018; Ghorbani et al., 2019) or its modification by creating a different objective function (Lian & Liu, 2019), or enabling of high learning rates through improved conditioning (Bjorck et al., 2018). Arora et al. (2019) demonstrate that (S)GD with BN is robust to the choice of the learning rate, with guaranteed asymptotic convergence, while a similar finding for GD with BN is made by Cai et al. (2019).

**Invariances in neural networks.** Cho & Lee (2017) propose optimizing over the Grassmann manifold using Riemannian GD. Liu et al. (2017) project weights and activations on the unit hypersphere and compute a function of the angle between them instead of inner products, and subsequently generalize these operators by scaling the angle (Liu et al., 2018). In (Li & Arora, 2020) the radial invariance is leveraged to prove that weight decay (WD) can be replaced by an exponential learning-rate scheduling for SGD with or without momentum. Arora et al. (2019) investigate the radial invariance and show that radius dynamics depends on the past gradients, offering an adaptive behavior to the learning rate. Here we go further and show that SGD projected on the unit hypersphere corresponds to Adam constrained to the hypersphere, and we give an accurate definition of this adaptive behavior.

**Effective learning rate.** Due to its scale invariance, BN can adaptively adjust the learning rate (van Laarhoven, 2017; Cho & Lee, 2017; Arora et al., 2019; Li & Arora, 2020). van Laarhoven (2017) shows that in BN-equipped networks, WD increases the effective learning rate by reducing the norm of the weights. Conversely, without WD, the norm grows unbounded (Soudry et al., 2018), decreasing the effective learning rate. Zhang et al. (2019) brings additional evidence supporting hypothesis in van Laarhoven (2017), while Hoffer et al. (2018a) finds an exact formulation of the effective learning rate for SGD in normalized networks. In contrast with prior work, we find generic definitions of the effective learning rate with exact expressions for SGD and Adam.

## 6 CONCLUSION

The spherical framework introduced in this study provides a powerful tool to analyse Adam optimization scheme through its projection on the $L_2$ unit hypersphere. It allows us to give a precise definition and expression of the effective learning rate for Adam, to relate SGD to a variant of Adam, and to identify geometric phenomena which empirically impact training. The framework also brings light to existing variations of Adam, such as $L_2$-regularization decoupling. This approach could be extended to other invariances in CNNs such as as filter permutation.

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
