# OpenReview forum: "A spherical analysis of Adam with Batch Normalization"
_ICLR.cc/2021/Conference — Reject_

### Official Review · AnonReviewer3 · 2020-10-20
**The analysis might be not sufficient**

**Rating:** 5
**Confidence:** 4

**Review:**

The paper attempts to analyze Adam with BN from a spherical perspective.

1. Though the authors derived the effective learning rate for Adam in Eq. 13, the analysis still needs to be more thorough. For example, how this effective learning rate is affected by the hyperparameters (i.e. learning rate, batch size). And in Eq. 13, since this term is complicated, involving ck, uk and b, a detailed and intuitive analysis should be provided.

2. About the second contribution, the paper claimed that SGD with BN behaves like a variant of Adam, AdamG*. However, this AdamG* has significant difference from original Adam: in AdamG* the division is by a *scalar* (not an element-wise division in Adam), this does not change the radially-invariant property of models. Thus, the connection between SGD with BN and AdamG* is obvious. I expect the authors provide some analysis on the connection between SGD with BN and AdamG*.

So I think is the contributions in this work are slightly below the acceptance bar of ICLR before these concerns are solved appropriately.

---

> ### Author Response · Authors · 2020-11-23
> **Response**
>
> We thank the reviewer for the comments.
>
> The purpose of this study is to provide an analysis of some underlying properties for the update of a network function that are due to the radial invariance, it is not to simplify the training dynamics.
>
> _‘Eq. 13, the analysis still needs to be more thorough. For example, how this effective learning rate is affected by the hyperparameters (i.e. learning rate, batch size)’_.
> Our analysis does not focus on the estimation of the training loss with sampled batches of data, so the batch size parameter is out of the scope. Regarding the analysis of effective quantities, they are difficult to control because we have little understanding of the highly-dimensional and non-convex loss landscape. Still, they highlight phenomena that, when iteratively neutralized, impact significantly optimization in terms of generalization (see Section 4). Furthermore, taking the analysis of the effective learning rate for SGD a notch higher, we highlight that in the presence of BN, SGD is actually an adaptive method (see Section 3).
>
> _‘In AdamG* the division is by a scalar (not an element-wise division in Adam), this does not change the radially-invariant property of models'_.
> This is a misunderstanding. The radial invariance is a property of models (which arises in presence of BN) regardless of the optimization method. Our theorem states that, for a given radially-invariant model, performing SGD is actually equivalent to performing a second-order moment adaptive method: AdamG*. In AdamG*, the division is by a scalar for each group of radially-invariant parameters (i.e., filters in the case of CNN). Therefore, each filter is adapted individually by the optimization algorithm. Since the division is done over groups of parameters it is not a global scheduling for SGD.
>
> _'Thus, the connection between SGD with BN and AdamG* is obvious'_. In its natural formulation, SGD does not use a second-order moment. Therefore, the equivalence with a scheme that does use such a second-order moment is not obvious.

---

### Official Review · AnonReviewer4 · 2020-10-27
**Weaker results than claimed**

**Rating:** 4
**Confidence:** 4

**Review:**

This work studies optimization dynamics for neural network models that are scaling invariant with respect to parameters. A general formulation of optimization algorithms is considered, covering many widely used algorithms like SGD and Adam. The projected dynamics (to the unit sphere) is studied, and the effective learning rate and update direction on the unit sphere are derived. Focusing on the projected dynamics, the equivalence is built between SGD and a type of "Adam". Then, different factors in the Adam dynamics that can potentially influence the optimization performance are identified, and empirically studied.

The paper is overall clear and easy to follow. My major concern is that the results are weaker than that is claimed in the abstract. Specifically,
1. The image optimization on the hypersphere is just a projected dynamics, it is not a close dynamics with respect to u. The radial part, although does not contribute to the loss directly, still plays a role in the dynamics. In this sense, the projected dynamics is not simplified compared to the original one. The effective learning rate is not easy to control. Moreover, the loss function of u is still highly non-convex and hard to study. No evidence is shown that L(u) is simpler than L(x).
2. The adaptive gradient methods are adaptive in at least two ways: a) the learning rate is adaptive; b) the learning rate is different for each parameter. The AdamG* considered in the paper does not use element-wise learning rate. It is more like an SGD with a learning rate schedule, instead of an adaptive gradient algorithm. Hence, the sentence "performing SGD alone is actually equivalent to a variant of Adam constrained to the unit hypersphere" in the abstract is misleading and conveys over-optimistic information.

---

> ### Author Response · Authors · 2020-11-23
> **Response**
>
> We thank the reviewer for the comments.
>
> First, we want to clarify the starting point of our study. The purpose of the paper is not to simplify optimization, it is to highlight some underlying phenomena in optimization that are due to the radial invariance property of models. The goal of training is to find the best possible function encodable by the network. Due to radial invariance, the parameter space projected on the unit hypersphere is topologically closer to the functional space of the network than the full parameter space. It hints that looking at optimization behaviour on the unit hypersphere might be interesting.
>
> *‘The image optimization on the hypersphere is just projected dynamics, it is not a close dynamics with respect to u’*. Eq.13 gives the update and dynamics with respect to $\mathbf{u}_k$. One could compute the effective quantities at each step and use it to do the optimization on the $\mathbf{u}_k$. This analysis gives an interesting theoretical insight on training SGD with BN, showing it is equivalent to performing an adaptive optimization method (Section 3). It also highlights geometrical phenomena that can be changed to impact optimization (Section 4).
>
> Second, AdamG* is an adaptive method. _‘The AdamG* considered in the paper does not use element-wise learning rate’_. This is a misunderstanding. Unlike Adam, which is adaptive by the second-order moment for each parameter, AdamG* is adaptive for each group of radially invariant parameters (filters for CNNs with BN). This scheme is very close to AdamG from Cho & Lee 2017, presented in the original paper as a variant of Adam.
>
> *‘ It is more like an SGD with a learning rate schedule’*. Since the division is done over groups of parameters, it is not a global scheduling for SGD.

---

### Official Review · AnonReviewer2 · 2020-10-29
**Interesting perspective but unclear consequences**

**Rating:** 5
**Confidence:** 4

**Review:**

Post-response update: I thank the authors for their response. I updated my score, but still think the paper needs improvement to be of interest to the ICLR community.

---


The submission analyzes the behavior of gradient descent and adaptive variants for scale-invariant models, including batch-normalization, by looking at the trajectory of the iterates when projected on the unit sphere. It contributes a formula for the equivalent learning rate if the gradient step was to be taken on the unit sphere for SGD and Adam and shows an approximate equivalence between gradient steps and normalized gradient steps taken on the unit sphere.

The spherical perspective is an interesting and insufficiently explored aspect of modern machine learning models. As the magnitude of the weights is essentially an irrelevant free parameter, understanding its behavior is likely to yield insights on how we train those overparametrized models.

The submission, however, does not provide a strong motivation for the presented results. The stated contribution that "in the presence of BN layers, standard SGD behaves like Adam without momentum" is also slightly over-selling the results of section 3. The main issue is that the significance of the explicit learning rates derived in section 2 or the equivalence between gradient descent and a normalized variant in section 3 is unclear. The manuscript does not provide sufficient context to understand what the derived formulas change about our theoretical understanding of those methods or what they imply for applications.

As it stands, I am worried that the submission would have little impact. I believe the manuscript would benefit from major revisions to be of interest to the community and my initial recommendation is a rejection.

The major issues that need to be adressed are:
- The technical definitions of the terms introduced in the work, like "equivalence of order 2", and the significance of the results, need to be stated in the main text. That two methods produce similar updates in terms of the overall model is interesting. But the significance of the result depends on what information it gives for the study or the application of the method, and this is currently unclear from the manuscript. For example;
  - why does the memory $v_k$ in the AdamG* update increase over time (with $\beta > 1$) if $\eta\lambda >2$? Or is it a setting that is not supported by the assumptions in Thm. 2?
  - What is the effect of L2 regularization if the complexity of the model is independent of the magnitude of the weight vector?
  - Why is $\beta = 1$ if there is no L2 regularization but $\beta < 1$ if $\eta\lambda < 2$?
  - Does it tell us anything about the difference between an Adagrad-style complete sum and Adam-style exponential moving average?
- The claimed contribution that SGD behaves like Adam without momentum is over-selling the results of section 3. Those results show that a gradient step is approximately a normalized gradient step projected on the unit ball. While this version might have similarities with Adam, this is not what the stated contribution would imply to most readers.
- The message of the paper is obscured by excessive formality. It might of course useful for a subset of readers to state the result in terms of topological equivalence to quotient manifolds, isomorphisms or to relate the results to the canonical metric on the sphere. But it makes the paper harder than necessary for the average reader, especially when technical terms, like the canonical metric, do not appear after their introduction. I strongly advise the authors to keep the main message of the paper accessible.

---

> ### Author Response · Authors · 2020-11-23
> **Response**
>
> We thank the reviewer for the comments.
>
> ### Significance of effective quantities
>
> First, we would like to stress that the effective quantities (effective learning rate and direction) are quantities used in the community and presented in a number of studies (van Laarhoven 2017; Hoffer et al 2018; Zhang et al 2019). Since we are interested in the change of the network function when training, Eq.13 corresponds to the functional update of the network. In other words, when training with BN and with a given optimization algorithm, the intrinsic update is in fact described in Eq. 13. The question is then: what are the theoretical insights provided by these quantities ?
>
> The fact is that the optimization of deep neural networks is poorly understood. The loss landscape has a high dimension and is non convex. Comparing steps of different optimization methods at different points of the parameter space will not give any fruitful insights for a better understanding of the optimization process. However, we can find equivalent schemes that will yield the same network function update during training (see Section 3), which reveals an unexpected property of SGD with BN. We can also study empirically geometrical phenomena highlighted by the framework, which turns out to have a significant impact on training of CNNs with BN on complex images domains (see Section 4). In other words, the modification of effective quantities leads to new optimization schemes that have different impacts in terms of speed and generalization.
>
> ### Equivalence theorem
>
> The purpose of this theorem is to show that in the presence of BN Layers, for the same model, standard SGD behaves like an adaptive method.
> This is not expected since SGD has no second-order moment in its update. Formally, we show the equivalence between SGD and the scheme dubbed AdamG*. AdamG* is an adaptive scheme relying on a second-order moment. Unlike Adam, which is adaptive w.r.t the second-order moment for each parameter, AdamG* is adaptive for each group of radially-invariant parameters (i.e., filters for CNNs with BN). This scheme is very close to AdamG from Cho & Lee 2017, presented in the original paper as a variant of Adam. Hence the link between SGD and a variant of Adam. The full sentence of this contribution in the original submission was "We show that, in presence of BN layers, SGD behaves like Adam without momentum, *adapted and constrained to the hypersphere.*".
>
> ### Details on equivalence theorem
>
> - *‘The technical definitions of the terms introduced in the work, like "equivalence of order 2", and the significance of the results, need to be stated in the main text’*. As stated in the paper just above the Theorem, we call « equivalent at order 2 in the step » a scheme equivalence that holds when we use for $r_k$ an expression that satisfies the radius dynamic with a Taylor expansion at order 2. This is the only assumption of the Theorem and it is verified in practice (see Appendix C.1.5 and Figure 5 in the supplementary material).
> - _‘Why does the memory $v\_k$ in the AdamG* update increase over time (with $\beta > 1$) if  $\eta \lambda > 2$ ? Or is it a setting not supported by Theorem 2 ?’_. Considering a radius approximated at order 2, the theorem holds for any positive values of $\lambda$, $\eta$ and $r_0$. It thus holds even if $\eta\lambda > 2$ (and hence $\beta > 1$), although this setting is not particularly practical. For standard values of hyperparameters $\lambda < 1$ (order of magnitude of 1e-4) and $\eta < 1$ (order of magnitude at most 1e-1), the higher-order terms of the radius in the Taylor expansion empirically become negligible and the equivalence stated in the theorem can be seen as a strong proxy to reality. Under such conditions, which are verified in practice, AdamG* is adapted by its order-two moments for each group of radially-invariant parameters.
> - *‘What is the effect of L2 regularization if the complexity of the model is independent of the magnitude of the weight vector ?’* We are not sure what "the complexity of the model" refers to, but the overall effect of L2 regularization in presence of BN is to exponentially schedule the learning rate. In this study, we go one step further by resolving the radius dynamic and demonstrate it contributes to the factor $\beta<1$ of second-order moment of the gradient norm, making SGD equivalent to an adaptive method.
> - *‘Why is $\beta=1$ if there is no L2 regularization but $\beta < 1$ if $\eta \lambda <  2$ ?'*. Again, these are mathematical conditions to have the equivalence. The expression of beta actually reveals that the L2 regularization parameters control the memory of the past gradients norm, which is interesting. If $\beta=1$ (no L2 regularization), then there is no attenuation (Adagrad-style); each gradient norm has the same contribution in the order-two moments. The effect of L2 regularization is to have a decay factor ($\beta < 1$) on past gradient norm in the order-2 moment.

---

### Author Response · Authors · 2020-11-23
**Paper revision**

Besides improving writing and fixing the typos in the paper, we made the following changes in the new revision:

- Clarification on the contribution of the equivalence between SGD and AdamG*, both in the introduction on the paper and in Section 3.
- Clarification on the motivation that drives this analysis in the introduction of Section 2.3.
- Precisions and interpretations below Theorem 4.

We also added an updated version of the Appendix.

---

### Decision · Program_Chairs · 2021-01-07
**Final Decision**

**Decision:**

Reject

**Comment:**

Three reviewers recommend rejecting or weak reject. The studied problem is interesting, but as one reviewer pointed out, it is not that clear how this work changes our theoretical understanding of those methods or what they imply for applications. Overall, I feel this work is on the borderline (probably it deserves higher score than the current score), but probably below the acceptance bar at the current form.